# Prediction model for in-hospital mortality in patients at high altitudes with ARDS due to COVID-19

**David Rene Rodriguez Lima**[1,2]*, **Cristhian Rubio Ramos**[1], **Andrés Felipe Yepes Velasco**[1‡], **Leonardo Andrés Gómez Cortes**[1‡], **Darío Isaías Pinilla Rojas**[1‡], **Ángela María Pinzón Rondón**[2‡], **Ángela María Ruíz Sternberg**[2‡]

**1** Critical and Intensive Care Medicine, Hospital Universitario Mayor-Méderi, Bogotá, Colombia, **2** Grupo de Investigación Clínica, Escuela de Medicina y Ciencias de la Salud, Universidad del Rosario, Bogotá, Colombia

☯ These authors contributed equally to this work.
‡ AFYV, LAGC, DIPR, ÁMPR and ÁMRS also contributed equally to this work.
* drrodriguezl@hotmail.com

**Data Availability Statement:** All relevant data are within the manuscript and its Supporting Information files.

## Abstract

## Introduction

The diagnosis of acute respiratory distress syndrome (ARDS) includes the ratio of pressure arterial oxygen and inspired oxygen fraction (P/F) $\leq$ 300, which is often adjusted in locations more than 1,000 meters above sea level (masl) due to hypobaric hypoxemia. The main objective of this study was to develop a prediction model for in-hospital mortality among patients with ARDS due to coronavirus disease 2019 (COVID-19) (C-ARDS) at 2,600 masl with easily available variables at patient admission and to compare its discrimination capacity with a second model using the P/F adjusted for this high altitude.

## Methods

This study was an analysis of data from patients with C-ARDS treated between March 2020 and July 2021 in a university hospital located in the city of Bogotá, Colombia, at 2,600 masl. Demographic and laboratory data were extracted from electronic records. For the prediction model, univariate analyses were performed to screen variables with p <0.25. Then, these variables were automatically selected with a backward stepwise approach with a significance level of 0.1. The interaction terms and fractional polynomials were also examined in the final model. Multiple imputation procedures and bootstraps were used to obtain the coefficients with the best external validation. In addition, total adjustment of the model and logistic regression diagnostics were performed. The same methodology was used to develop a second model with the P/F adjusted for altitude. Finally, the areas under the curve (AUCs) of the receiver operating characteristic (ROC) curves of the two models were compared.

**Funding:** The author(s) received no specific funding for this work.

**Competing interests:** The authors have declared that no competing interests exist.

## Results

A total of 2,210 subjects were included in the final analysis. The final model included 11 variables without interaction terms or nonlinear functions. The coefficients are presented excluding influential observations. The final equation for the model fit was g(x) = age (0.04819)+weight(0.00653)+height(-0.01856)+haemoglobin(-0.0916)+platelet count (-0.003614)+ creatinine(0.0958)+lactate dehydrogenase(0.001589)+sodium(-0.02298) +potassium(0.1574)+systolic pressure(-0.00308)+if moderate ARDS(0.628)+if severe ARDS(1.379), and the probability of in-hospital death was p (x) = e $^{g (x)}$/(1+ e $^{g (x)}$). The AUC of the ROC curve was 0.7601 (95% confidence interval (CI) 0.74–0, 78). The second model with the adjusted P/F presented an AUC of 0.754 (95% CI 0.73–0.77). No statistically significant difference was found between the AUC curves (p value = 0.6795).

## Conclusion

This study presents a prediction model for patients with C-ARDS at 2,600 masl with easily available admission variables for early stratification of in-hospital mortality risk. Adjusting the P/F for 2,600 masl did not improve the predictive capacity of the model. We do not recommend adjusting the P/F for altitude.

## Introduction

The most widely used definition of acute respiratory distress syndrome (ARDS) is the one proposed in 2012 by the Berlin consensus, which requires the presence of multilobar alveolar infiltrates in chest images, an identifiable cause related to ARDS and hypoxemia quantified by the ratio of pressure arterial oxygen and inspired oxygen fraction (P/F) $\leq$ 300 [1]. The Berlin definition also classifies the severity of ARDS based on the P/F: mild with P/F> 200 $\leq$ 300, moderate with P/F> 100 $\leq$ 200 and severe with P/F $\leq$ 100. The reported mortality rates are 27%, 32% and 45%, respectively [1]. In 2016, the Large Observational Study to Understand the Global Impact of Severe Acute Respiratory Failure (LUNG-SAFE) found a 10% incidence of ARDS among all patients admitted to intensive care, with mortality rates of 34.9%, 40.3% and 46.1% for mild, moderate and severe ARDS, respectively [2].

Barometric pressure (BP) decreases with altitude, leading to a decrease in arterial oxygen pressure (PaO$_2$), a phenomenon known as hypobaric hypoxemia [3], which produces lower P/F values [4]. With this theoretical consideration, the Berlin definition proposes an adjustment for the P/F according to altitudes beginning at 1,000 masl [1]:

$$P/F \text{ adjusted} = P/F \text{ measured} \times (BP \text{ of altitude} \div 760) \tag{1}$$

More than 520 million people worldwide live in areas more than 2,500 masl, and many of these live in the Andean Latin American region [5]. This recommendation to measure their P/F differently than individuals who live at altitudes less than 1,000 masl has been made with no evidence from clinical studies; furthermore, it is unknown whether the height-adjusted P/F corresponds to the mortality predicted by the Berlin definition [1] and LUNG-SAFE for the different degrees of severity of ARDS [2]. An observational study among 249 patients suggested that the P/F adjustment is inaccurate for patients receiving invasive mechanical ventilation (IMV) who are acclimatized to altitudes > 1,500 masl and is not related to mortality [6]. Even for critically ill patients with COVID-19 who were treated at such altitudes, the

adjustment of P/F was not taken into account when diagnosing ARDS or its severity due to the lack of clinical evidence [7].

Hypobaric hypoxemia at high altitude has been associated with increased mortality in some conditions, such as influenza pneumonia [8], and in others, such as pulmonary tuberculosis, it has been shown to be a protective factor [9, 10]. However, there is little evidence of the impact on mortality among patients with ARDS [11] or COVID-19 [12].

COVID-19 can cause death and disability by multiple mechanisms; however, most deaths occur in the presence of hypoxemia due to viral pneumonia [13, 14]. These patients are managed with different forms of ventilatory support, which means that the majority of patients with respiratory failure due to COVID-19 meet the criteria for ARDS using the Berlin definition [14]. Although ARDS has its own pathophysiological characteristics [15], the behavior of ARDS in patients with COVID-19 seems to be similar to that of other causes [4, 16].

The initial step in reducing mortality among patients with ARDS is the early identification of risk factors [17, 18]; for this reason, several prediction models have been developed for unfavorable outcomes in patients with ARDS [19–21] and in specific populations, such as rheumatology patients with COVID-19 [22] and patients with pancreatitis [23]. However, none of these include patients who live above 1,000 masl.

The objective of this work was to develop and compare two prediction models for in-hospital mortality among patients with C-ARDS, the first with the altitude-adjusted P/F and the second with the unadjusted P/F, to determine if the P/F adjustment improves the predictive capacity and better discriminates high-risk patients.

## Methods

The implementation of this protocol was approved by the ethics committee of the Universidad del Rosario (DVO005 1802-CV1489). This study did not receive funding for its development. Each patient or legal representative signed a written general institutional informed consent form upon admission to the Hospital Universitario Mayor Méderi, in which he or she expressed his or her will to allow the use of clinical history data, diagnostic images and the publication of this information for academic purposes. This consent was physically filed in the patient's medical record.

All patients who met the inclusion criteria from March 19, 2020, to July 31, 2021, were included. Electronic records were accessed for research purposes, and all data were collected from December 10, 2021, to June 30, 2022. The inclusion criteria were patients with 1) a positive swab for severe acute respiratory syndrome coronavirus 2 (SARS-CoV-2); 2) diffuse bilateral alveolar infiltrates on chest X-ray or tomography; 3) a P/F <300, independent of the oxygen administration system; and 4) no clinical evidence of left atrial hypertension. The definition of ARDS that was used was the Berlin definition [1], with the Kigali modification [24]. Therefore, patients who met the ARDS criteria were included regardless of whether they were on IMV. Patients were excluded if 1) they had chronic lung disease with dependence on supplemental oxygen at home prior to admission; 2) no informed consent was obtained; or 3) the reason for hospital admission was a condition other than COVID-19.

### Data extraction

The data were extracted from the electronic records of the Mayor Méderi University Hospital, which is in the city of Bogotá, Colombia, at 2,640 masl. During the study period, 7,345 patients with confirmed COVID-19 were treated at the hospital, and the hospital expanded its 99 intensive care beds to 131 beds during this phase of the pandemic. The variable data were recorded in a file called the "dictionary of variables". The following variables were extracted: sex, age,

weight, height, leukocyte count, neutrophil count, lymphocyte count, monocyte count, eosinophil count, basophil count, hemoglobin, hematocrit, platelet count, neutrophil/lymphocyte ratio (N/L ratio), creatinine, d-dimer, lactate dehydrogenase, C-reactive protein, sodium, potassium, body mass index, ferritin, procalcitonin, chlorine, calcium, glucose, albumin, alkaline phosphatase, aspartate aminotransferase, alanine aminotransferase, respiratory rate, heart rate, systolic blood pressure, diastolic blood pressure, Charlson comorbidity index, P/F at the time of ARDS diagnosis, P/F at admission, worst P/F during hospitalization, and use of IMV. Except for the P/F values, all measurements, including laboratory data and physiological findings, were taken within the first 24 hours of admission. The P/F at admission was the first P/F measured during hospitalization, the P/F at diagnosis was the P/F closest to the time of the chest X-ray or tomography that showed ARDS, and the worst P/F was the lowest throughout hospitalization. The database was coded so that no patient could be identified before the analyses were performed.

The main outcome evaluated was in-hospital mortality. For the development of the P/F model without adjusting for ARDS, ARDS was classified as mild with a P/F> 200 ≤ 300, moderate with a P/F> 100 ≤ 200 and severe with a P/F ≤ 100.

For the development of the model with the adjusted P/F, Eq 1 was used, considering that the city of Bogotá is located at 2,600 masl, with a barometric pressure of 560 millimeters of mercury (mmHg), and an adjustment factor of 0.74 (560 mmHg/760 mmHg) was used. Based on this adjustment, patients with a P/F> 148 ≤ 222 were considered to have mild ARDS, those with a P/F> 74 ≤ 148 moderate ARDS, and those with a P/F ≤ 74 severe ARDS.

In the final model, 30 variables were expected to be evaluated, calculating 20 events (deaths) for each variable entered in the initial model. In the model, with an expected mortality of 50%, a minimum sample size of 1,200 patients was calculated.

## Statistical analysis

Two prediction models (unadjusted P/F and adjusted P/F) were developed using the same methodology.

Univariate logistic regression models were used to evaluate which variables were associated with in-hospital mortality among patients with C-ARDS. The dependent variable was a binary result with "1" indicating death and "0" indicating survival. The independent variables were categorized into continuous variables or categorical variables. For continuous variables, odds ratios (ORs) are reported for each unit of increase in the parameter. For categorical variables, the ORs are reported for each category in relation to the reference category.

The AUC of the univariate model of P/F is presented graphically to evaluate its discrimination capacity as the only independent variable.

Variables with p <0.25 in the univariate analyses were included for the selection of covariates in the initial model. In addition, variables with a prevalence <10% or with more than 10% of observations missing were excluded from the initial model.

For the initial model, automatic selection was performed with the backward stepwise method, and the variables with p <0.1 were considered to be independently associated with mortality. The automatic selection finally generated a main effects model. To make use of all available patients and improve statistical power, the data imputation technique by stochastic regression was used. The coefficients of the observations were reported based on complete data and on the imputed data.

In addition, to avoid overfitting, bootstrapping was performed to adjust the estimated coefficients in the main effects model. The bootstrapping technique used resampling with replacement, and the sampling was repeated 500 times. The coefficients estimated by this technique reduce the estimated coefficient but provide a better prediction for future samples.

The potential interactions between the variables included in the main effects model were tested by adding them one by one. The interaction terms with p <0.1 were considered for inclusion in the model. The linearity of the variables with their logarithmic scale is essential for the fit of the model. To carry out this evaluation, multivariable fractional polynomials (MFPs) were used to test whether other power terms were superior to the linear term.

The fit of the model was assessed from two aspects: a summary measure and regression diagnostics. The model discrimination was graphically represented by the AUC. The goodness of fit was evaluated using the Hosmer–Lemeshow test, which was calculated using Pearson's chi-square statistic from the g × 2 table of the values of the observed and estimated frequencies. The variable g refers to the number of groups. The level of statistical significance p <0.05 indicated that the model was significantly different from the observed result.

The logistic regression diagnostics were calculated to see if the model fit in the whole group of covariate patterns. Pearson's residuals and marginal residuals were plotted. To evaluate the outliers, the studentized residuals were graphed, and for leverage, the hat values were evaluated. Influential observations were understood as the product of leverage and outliers. Cook's distances are presented graphically as a summary measure of influence. To evaluate the influence, the influence plot of the car package was used in RStudio. The coefficients are presented in the final model, excluding the influencing data.

This model construction process was performed for all patients in the cohort with adjusted and unadjusted P/F. Finally, the two models are presented with their summary measures and evaluation. All statistical analyses were performed in RStudio version 4.1

## Results

At Mayor Méderi University Hospital, 7,345 patients with a confirmed diagnosis of COVID-19 were treated between March 1, 2020, and July 31, 2022. Fifty-six patients were excluded from this study for not having a positive swab for SARS-CoV-2. A total of 4,928 patients were excluded because they did not have a chest image compatible with ARDS, and 943 did not have a P/F ≤ 300.

The diagnosis of C-ARDS was confirmed in 2,313 patients; however, in 103 patients, the P/F at the time of ARDS diagnosis could not be established, only the worst P/F during hospitalization. Thus, 2,210 patients were included in the final analysis.

Without adjusting the P/F, 37.7% (n = 833), 42.3% (n = 948), and 19.1% (n = 429) patients presented mild, moderate, and severe C-ARDS, respectively. The overall mortality was 43.7% (n = 965); in mild cases, it was 29.7% (n = 247), 46.3% (n = 439) in moderate cases and 65% (n = 279) in severe cases (Fig 1A).

When the P/F was adjusted, 168 patients did not meet the criteria for ARDS due to oxygenation (P/F of 223 to 300). This decreased the incidence of C-ARDS by 7.6% in this cohort, leaving 2,042 patients for analysis. With the adjusted P/F, 50.4% (n = 1030), 38.1% (n = 778), and 11.5% (n = 234) patients presented mild, moderate, and severe C-ARDS, respectively. The overall mortality was 46.5% (n = 949); in mild cases, it was 36.2% (n = 373), 52.6% (n = 409) in moderate cases and 71.3% (n = 167) in severe cases (Fig 1B). Among patients who did not meet the criteria for ARDS when the P/F was adjusted, mortality was 9.5% (n = 16).

To begin the analysis, the predictive capacity was evaluated in a univariate analysis of the adjusted and unadjusted P/F for in-hospital mortality. Discrimination of the P/F assessed by the AUC with the unadjusted P/F was 0.6552 (95% CI: 0.6322–0.6783) (Fig 2A), and the AUC of the adjusted P/F was 0.6484 (95% CI: 0.6247–0.6721) (Fig 2B). There was no statistically significant difference between the curves (p value = 0.6855 evaluated through De Long's test).

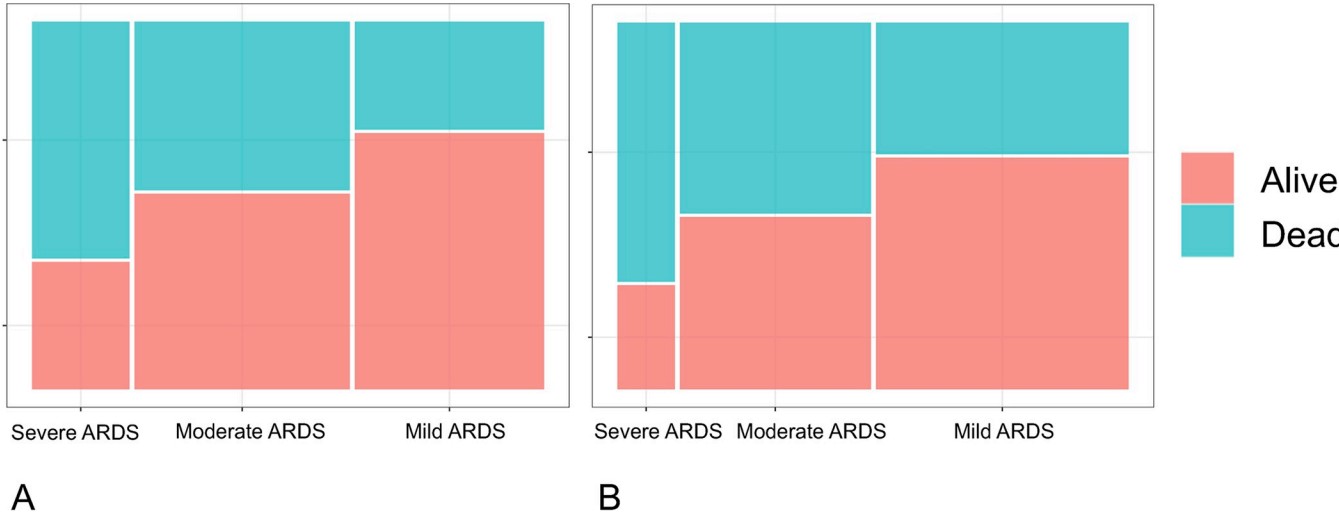

**Fig 1. Mortality for each level of C-ARDS severity.** A. P/F not adjusted. B. P/F adjusted.

The development of the prediction model began with the unadjusted P/F. The univariate logistic regression analyses are presented in Table 1.

Variables with less than 10% of missing data and with p <0.25 were selected for the initial model. Age, weight, height, lymphocyte count, basophil count, hemoglobin, hematocrit, platelet count, N/L ratio, creatinine, lactate dehydrogenase, sodium, potassium, heart rate, systolic pressure, diastolic pressure, Charlson comorbidity index, P/F at diagnosis, sex, and severity of ARDS were considered candidates for inclusion in the initial model (Table 1).

In the complete model, hematocrit was excluded since it measured the same variable as hemoglobin. Lymphocyte count was included, not the N/L ratio, since the significance of the relationship was evidenced by the lymphocyte count. ARDS stratified by severity was selected, not the P/F at diagnosis, since both variables measured oxygenation.

In total, 1,942 subjects had complete data, and the selection of the variables in the final model was performed automatically using the backward stepwise strategy. Variables with p

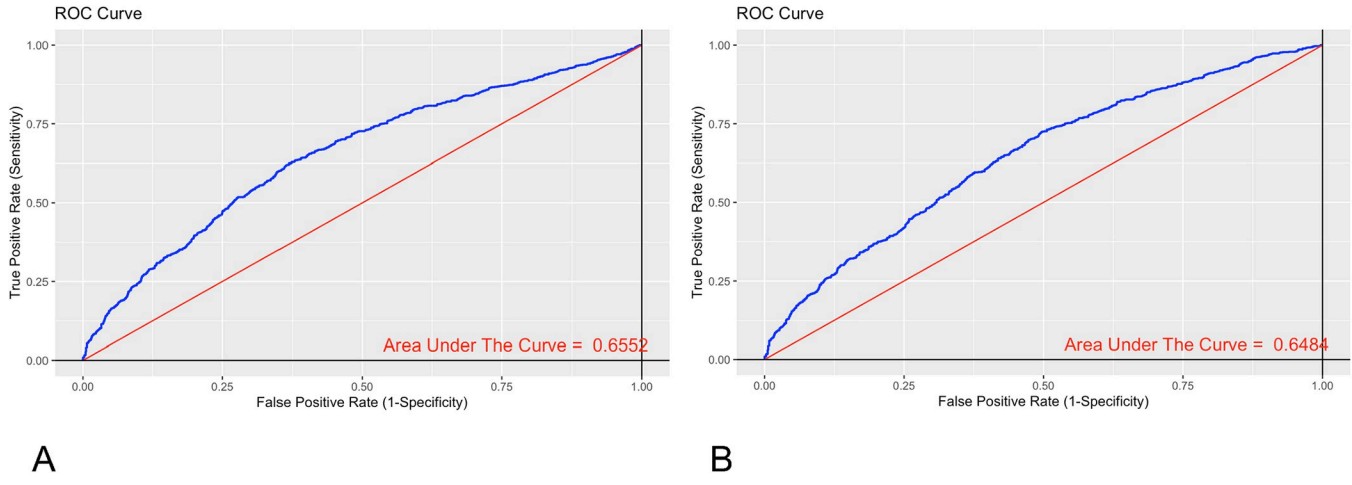

**Fig 2. ROC curves of P/F for in-hospital mortality in a univariate model.** A. P/F not adjusted. B. P/F adjusted.

**Table 1. Univariate logistic regression analyses.**

| Variable | OR | 95% CI | P | Missing | (%) |
|---|---|---|---|---|---|
| Age | 1.0474 | 1.0402–1.0546 | 0.000 | 0 | 0 |
| Weight | 0.9881 | 0.9823–0.9940 | 0.000 | 0 | 0 |
| Height | 0.9705 | 0.9608–0.9802 | 0.000 | 0 | 0 |
| Leukocyte count | 1.0034 | 0.9916–1.0153 | 0.574 | 12 | 0.54 |
| Neutrophil count | 1.0016 | 0.9829–1.0207 | 0.864 | 13 | 0.58 |
| Lymphocyte count | 0.9093 | 0.8192–1.0093 | 0.074 | 12 | 0.54 |
| Monocyte count | 1.0537 | 0.8654–1.2828 | 0.603 | 12 | 0.54 |
| Eosinophil count | 1.1238 | 0.3491–3.6180 | 0.845 | 12 | 0.54 |
| Basophil count | 0.0948 | 0.0028–3.1917 | 0.189 | 12 | 0.54 |
| Hematocrit | 0.9769 | 0.9649–0.9890 | 0.000 | 11 | 0.49 |
| Hemoglobin | 0.9158 | 0.8836–0.9492 | 0.000 | 11 | 0.49 |
| Platelet count | 0.9969 | 0.9959–0.9959 | 0.000 | 17 | 0.76 |
| Neutrophil/lymphocyte ratio | 1.0225 | 1.0136–1.0315 | 0.000 | 13 | 0.58 |
| Creatinine | 1.1350 | 1.0796–1.1933 | 0.000 | 20 | 0.90 |
| D-dimer | 1.0000 | 1.0000–1.0000 | 0.275 | 133 | 6.0 |
| Lactate dehydrogenase | 1.0015 | 1.0010–1.0019 | 0.000 | 86 | 3.89 |
| C-reactive protein | 1.0001 | 0.9993–1.0010 | 0.749 | 198 | 8.9 |
| Sodium | 0.9740 | 0.9581–09901 | 0.002 | 184 | 8.32 |
| Potassium | 1.2624 | 1.1126–1.4324 | 0.000 | 181 | 8.19 |
| Respiratory rate | 1.0086 | 0.9892–1.0284 | 0.386 | 0 | 0 |
| Heart rate | 0.9945 | 0.9901–0.9989 | 0.014 | 0 | 0 |
| Systolic pressure | 0.9970 | 0.9933–1.0008 | 0.124 | 0 | 0 |
| Diastolic pressure | 0.9896 | 0.9830–0.9962 | 0.002 | 0 | 0 |
| Body mass index | 0.9962 | 0.9804–1.0123 | 0.644 | 0 | 0 |
| Charlson comorbidity index | 1.4291 | 1.3561–1.5272 | 0.000 | 1 | 0.04 |
| P/F diagnosis | 0.9917 | 0.9904–0.9931 | 0.000 | 0 | 0 |
| Female sex | Reference: | | | 0 | 0 |
| Male sex | 0.7682 | 0.6433–0.9173 | 0.004 | 0 | 0 |
| Mild ARDS | Reference: | | | 0 | 0 |
| Moderate ARDS | 2.0462 | 1.6820–2.4892 | 0.000 | 0 | 0 |
| Severe ARDS | 4.4128 | 3.4437–5.6546 | | 0 | 0 |
| High blood pressure | 1.0660 | 0.8792–1.2925 | 0.516 | 0 | 0 |
| Variables with more than 10% of missing data | | | | | |
| Ferritin | 1.0002 | 1.0001–1.0003 | 0.002 | 1720 | 77.8 |
| Procalcitonin | 1.0402 | 1.0402–1.0626 | 0.000 | 1122 | 50.7 |
| Chlorine | 0.9831 | 0.9677–0.9987 | 0.034 | 296 | 13.39 |
| Calcium | 0.5398 | 0.4513–0.6458 | 0.000 | 866 | 39.18 |
| Glucose | 1.0021 | 1.0008–1.0035 | 0.002 | 1178 | 53.3 |
| Albumin | 0.2788 | 0.0954–0.8148 | 0.020 | 2127 | 96.2 |
| Alkaline phosphatase | 1.0006 | 0.9993–1.0019 | 0.347 | 1723 | 77.9 |
| Alanine aminotransferase | 1.0003 | 0.9997–1.0008 | 0.355 | 237 | 10.7 |
| Aspartate aminotransferase | 1.0009 | 1.0001–1.0016 | 0.018 | 238 | 10.7 |
| Total bilirubin | 1.1158 | 0.9800–1.2704 | 0.098 | 243 | 10.9 |
| Variables not considered for entry into the model | | | | | |
| P/F at admission | 0.9938 | 0.9926–0.9950 | 0.000 | 0 | 0 |
| Lower P/F | 0.9792 | 0.9769–0.9815 | 0.000 | 0 | 0 |
| Invasive mechanical ventilation | 6.1955 | 5.1446–7.4610 | 0.000 | 0 | 0 |

Univariate analysis by mortality. P/F unadjusted.

**Table 2. Final model after stepwise selection of the covariates.**

| Covariate | Imputed data model | Model with complete data[a] | Bootstrapping |
|---|---|---|---|
| Age | 0.0469 (0.0388, 0.0549) | 0.0434 (0.0350, 0.0519) | 0.0434 (0.0376, 0.0544) |
| Weight | 0.0060 (-0.0018, 0.0137) | 0.0075 (-0.0007, 0.0157) | 0.0074 (-0.0017, 0.0130) |
| Height | -0.0181 (-0.0304, -0.0059) | -0.0173 (-0.0305, -0.0042) | -0.0173 (-0.0300, -0.0061) |
| Hemoglobin | -0.0879 (-0.1326, -0.0431) | -0.0742 (-0.1208, -0.0277) | -0.0742 (-0.1313, -0.0404) |
| Platelet count | -0.0035 (-0.0045, -0.0024) | -0.0036 (-0.0047, -0.0025) | -0.0035 (-0.0046, -0.0023) |
| Creatinine | 0.0942 (0.0392, 0.1491) | 0.0929 (0.0349, 0.1509) | 0.0929 (0.0349, 0.1524) |
| Lactate dehydrogenase | 0.0013 (0.0008, 0.0018) | 0.0013 (0.0008, 0.0018) | 0.0012 (0.0006, 0.0019) |
| Sodium | -0.0181 (-0.0355, -0.0007) | -0.0231 (-0.0420, -0.0042) | -0.0230 (-0.0372, 0.0013) |
| Potassium | 0.1618 (0.0150, 0.3086) | 0.1824 (0.0279, 0.3368) | 0.1823 (0.0193, 0.3063) |
| Systolic pressure | -0.0032 (-0.0074, 0.0011) | -0.0039 (-0.0084, 0.0005) | -0.0039 (-0.0072, 0.0007) |
| Moderate ARDS | 0.6586 (0.4409, 0.8763) | 0.5790 (0.3494, 0.8085) | 0.5789 (0.4197, 0.8769) |
| Severe ARDS | 1.4171 (1.1376, 1.6966) | 1.2489 (0.9582, 1.5396) | 1.2489 (1.104, 1.678) |

The final model coefficients with their 95% CIs are reported with complete data after data imputation and bootstrapping.

[a] 1,942 patients with complete data.

<0.1 were included in the final analysis to select the variables independently associated with mortality. Automatic selection generated a final model.

The final model included 11 variables: age, weight, height, hemoglobin, platelet count, creatinine, lactate dehydrogenase, sodium, potassium, systolic pressure, and severity of ARDS. Data imputation was performed by stochastic regression, and 2,210 patients were included in the model. Table 2 shows the coefficients with their 95% confidence intervals (CIs) both for patients with complete data and after data imputation.

Bootstrapping was performed, and the coefficients remained unchanged compared to the original analysis with and without data imputation (Table 2).

The interaction terms were evaluated for all plausible interactions, and no statistically significant interactions were found between the variables. The assumption of linearity for continuous variables was evaluated in the final model by using MFPs, which showed that other nonlinear functions were no better than linear functions. As a result, we adopted the original model as the final model.

To evaluate the model, Pearson's residuals were plotted (Fig 3), and the residuals against the predictors were observed one by one. The trend of the relationship was linear for all variables. This indicated that the model was adequately specified and that it was not necessary to add an additional predictor.

The marginal residuals were also graphed, the response variable, mortality, was graphed against the explanatory variables, and the observed data and the model predictions are shown in blue–red in Fig 4. An adequate adjustment was made for most of the variables, except platelet count; however, this did not improve with conversion, so it remained in its original form.

To evaluate the outliers, the studentized residuals were evaluated, and to evaluate the leverage, the hat values were evaluated. Influential observations can be understood as the product of leverage and outliers. Cook's distances are a summary measure of influence (Fig 5). The sizes of the resulting circles were proportional to Cook's distances. Four highly influential observations were obtained: 7,43,1367,1559.

The highly influential measures of the model were excluded, and the coefficients were estimated (Table 3). Discrimination was evaluated with the AUC of the ROC curve without influencing data. This was acceptable, with an AUC of 0.7601 (95% CI 0.74–0.78) (Fig 6A).

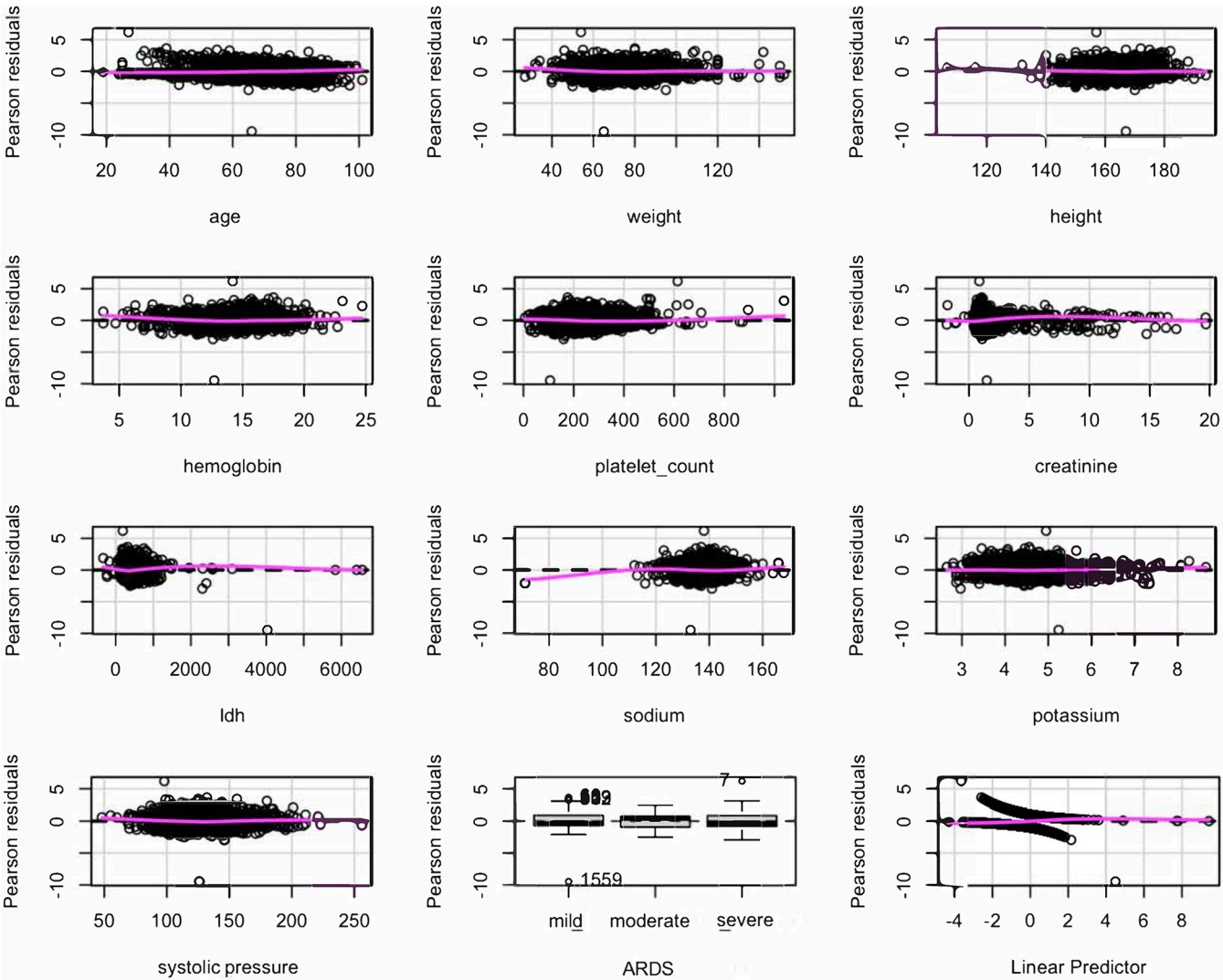

**Fig 3. Pearson's residuals.**

The calibration of the model without influencing data was evaluated using the Hosmer–Lemeshow goodness-of-fit test with a chi-squared = 3.6418, df = 8 and p value = 0.8879 and indicated no significant difference between the predicted probabilities and the observed probabilities. Fig 7A shows the calibration of the model.

Finally, we used the model without influencing data to predict the in-hospital mortality among patients with C-ARDS (Eq 2):

$$p\left(x\right) = \frac{e^{g(x)}}{1 + e^{g(x)}} \tag{2}$$

where

$$
\begin{aligned}
g(x) = {} & \text{age}(0.04819) + \text{weight}(0.00653) + \text{height}(-0.01856) + \text{haemoglobin}(-0.0916) \\
& + \text{platelet count}(-0.003614) + \text{creatinine}(0.0958) \\
& + \text{lactate dehydrogenase}(0.001589) + \text{sodium}(-0.02298) + \text{potassium}(0.1574) \\
& + \text{systolic pressure}(-0.00308) + \text{if moderate ARDS}(0.628) \\
& + \text{if severe ARDS}(1.379)
\end{aligned}
\tag{3}
$$

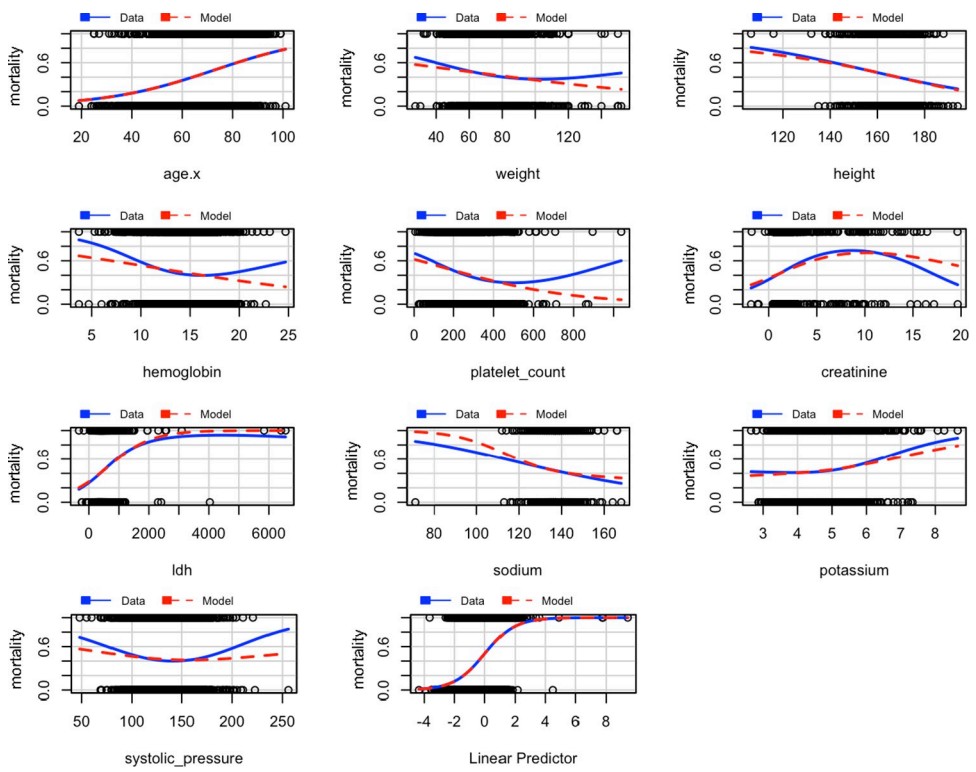

**Fig 4. Marginal residuals.**

With the same construction strategy of the unadjusted P/F model, a model with the adjusted P/F was developed. With the adjusted P/F criterion, 2,042 patients were used to develop the model. In the S1 File, Table 1 shows the univariate analyses for mortality, and Table 2 shows the coefficients of the variables selected for inclusion in the model by means of the backward stepwise technique for the complete data, the imputed data and bootstrapping. The interaction

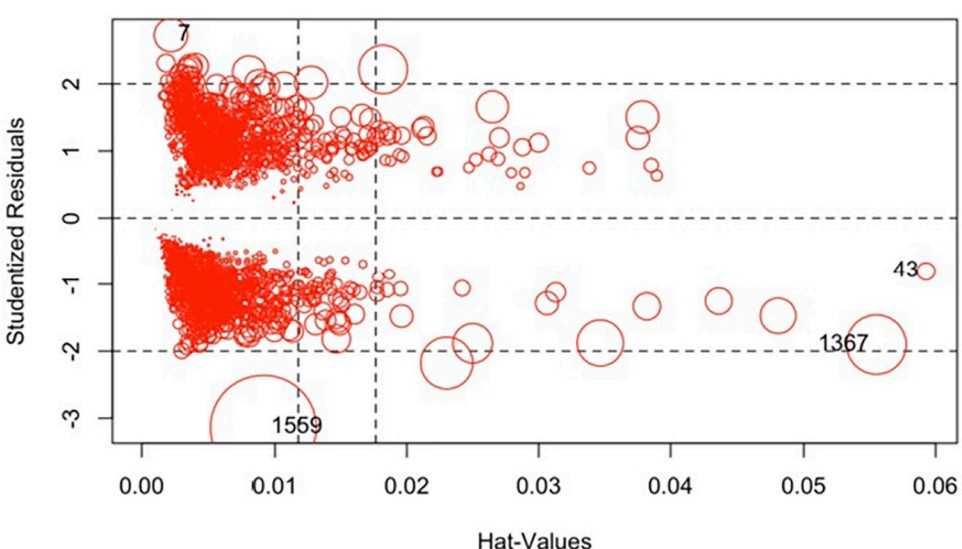

**Fig 5. Graph of studentized residuals versus hat values.**

**Table 3. Model coefficients with imputed data and without influencing data.**

| Covariate | Imputed data model | Model without influencing data |
|---|---|---|
| Age | 0.04685 | 0.04819 |
| Weight | 0.00596 | 0.00653 |
| Height | -0.01812 | -0.01856 |
| Hemoglobin | -0.0879 | -0.0916 |
| Platelet count | -0.003497 | -0.003614 |
| Creatinine | 0.0942 | 0.0958 |
| Lactate dehydrogenase | 0.001319 | 0.001589 |
| Sodium | -0.01812 | -0.02298 |
| Potassium | 0.1618 | 0.1574 |
| Systolic pressure | -0.00317 | -0.00308 |
| Moderate ARDS | 0.659 | 0.628 |
| Severe ARDS | 1.417 | 1.379 |

terms were evaluated for all plausible interactions, and no statistically significant interactions were found among the variables. The assumption of linearity for continuous variables was evaluated in the final model by using MFPs, which showed that other nonlinear functions were no better than linear functions. The presence of influential data was evaluated (Fig 1 of the S1 File), and four observations were excluded: 5, 39, 1253 and 1441. In the final model, there were nine variables. Table 4 shows the coefficients of the final model with imputed data and without influencing data.

Discrimination of the model with the adjusted P/F was evaluated with the AUC of the ROC curve without influencing data. This was acceptable, with an AUC of 0.754 (95% CI 0.73–0.77) (Fig 6B).

The calibration of the model with the adjusted P/F without influencing data was evaluated using the Hosmer–Lemeshow goodness-of-fit test with a chi-squared = 1.8106, df = 8 and p value = 0.9863, indicating that there was no significant difference between the predicted and observed probabilities. Fig 7B shows the calibration of this model.

The AUC of the ROC curve of the model was compared with those of the unadjusted and adjusted P/F models, and there was no statistically significant difference between the curves (p value = 0.6795) evaluated with De Long's test.

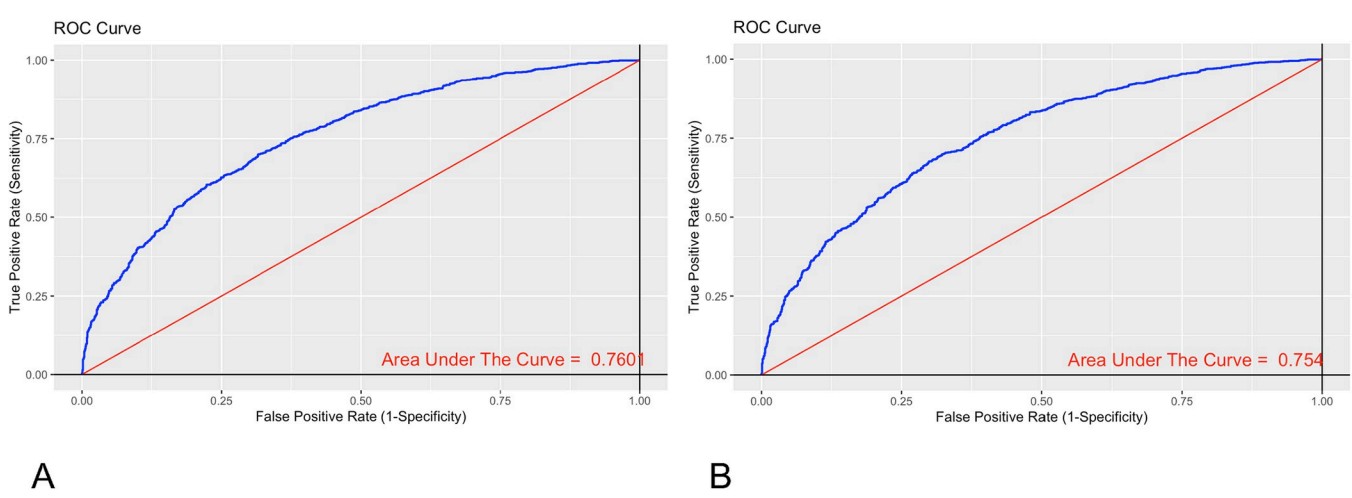

**Fig 6. ROC curve of the model without influencing data.** A. P/F not adjusted. B. P/F adjusted.

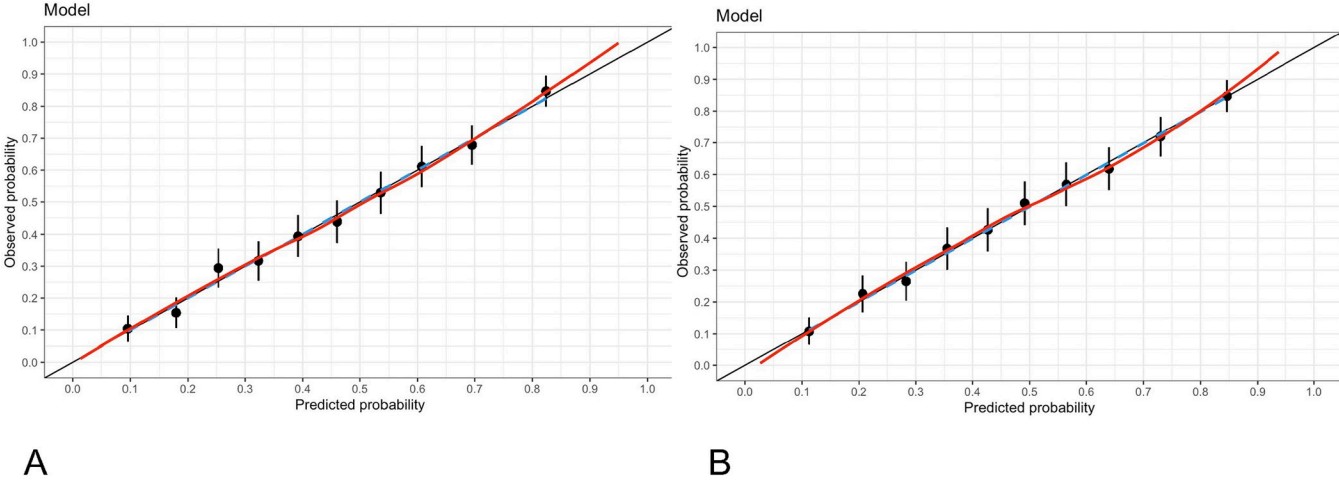

**Fig 7. Model calibration graph.** A. P/F not adjusted. B. P/F adjusted.

## Discussion

The prediction of mortality among critically ill patients has been studied extensively in recent years, and some studies have evaluated the predictive capacity of the Acute Physiology and Chronic Health Evaluation (APACHE) IV, Sequential Organ Failure Assessment (SOFA) and APACHE II in patients with ARDS, which have had poor to moderate discrimination in these patients [25, 26]. This makes it necessary to develop predictive models adjusted for ARDS patients. Zhao et al. [27] developed a model based on APACHE III, age, surfactant protein D and interleukin (IL)-8 based on the cohort of the Assessment of Low tidal Volume and elevated End-expiratory volume to Obviate Lung Injury (ALVEOLI) study with external validation in the Fluid and Catheter Treatment Trial (FACTT) and Validating Acute Lung Injury biomarkers for Diagnosis (VALID) studies and found a moderate discrimination of the model; however, the use of surfactant protein D and IL-8 is not routinely practiced in intensive care units, making their application difficult in daily practice. There are other prediction models for ARDS, such as that of Zhang et al. [21] in which a model with eight variables was presented with a good discrimination capacity and an AUC of 0.85, emphasizing that the mortality of the patients in the study was lower (21%) than that reported in other ARDS cohorts; this may be because the patients were enrolled in randomized clinical trials and received specific interventions, which reduced the guarantee of external validity. Recently, Ye et al. [28] developed a prediction model with nine variables for patients with ARDS receiving IMV and found an AUC of 0.75, similar to that shown in our model.

Without adjusting the P/F for altitude, overall mortality in this cohort was 43.7%, mortality for mild ARDS was 29.7%, 46.3% for moderate ARDS and 65% for severe ARDS. When adjusting the P/F, the incidence of ARDS decreased by 7.6%, and the overall mortality was 46.5%; in mild cases, it was 36.2%, 52.6% in moderate cases and 71.3% in severe cases. Mortality was higher than that reported in other studies on ARDS [1, 2, 16, 29, 30], which may be explained by the age of patients in this cohort (a median of 68 years, IQR 58–77) and that they were managed during the COVID-19 outbreak before mass vaccination was available (Table 5).

In this study, two prediction models were developed for hospital mortality among patients with C-ARDS, the first without adjustment of the P/F and the second with the P/F adjusted for altitude. The first final model included 11 variables (Table 3), and the second included 9 variables (Table 4). Discrimination was acceptable and similar in the first and second models, as

**Table 4. Model coefficients using the adjusted P/F with imputed data and without influencing data.**

| Covariate | Imputed data model | Model without influencing data |
|---|---|---|
| Age | 0.04736 | 0.04853 |
| Height | -0.01268 | -0.01271 |
| Hemoglobin | -0.0711 | -0.0747 |
| Platelet count | -0.003386 | -0.003527 |
| Creatinine | 0.0972 | 0.0979 |
| Lactate dehydrogenase | 0.001191 | 0.001441 |
| Sodium | -0.01984 | -0.02522 |
| Potassium | 0.1879 | 0.1854 |
| Moderate ARDS | 0.710 | 0.696 |
| Severe ARDS | 1.387 | 1.361 |

reflected in the AUCs of the ROC curves of 0.76 and 0.75, respectively (Fig 6A and 6B). The parameters incorporated in the two models were collected at hospital admission and the P/F at the time of C-ARDS diagnosis, which allowed early recognition of patients at high risk of in-hospital death. This is also the first prediction model described in the literature to consider the recommendation of the Berlin consensus that the P/F be adjusted for altitude. However, this adjustment did not improve the predictive capacity of the model either in the univariate analysis or the analysis adjusted for covariates. Furthermore, if patients who have a lower risk of mortality (9.5%), but a high probability of poor clinical outcomes are excluded, they miss the benefits of early detection.

In a study published by the same research group as this work, it was demonstrated that at an altitude of 2640 masl, there exists a significant association between lower P/F ratios and in-hospital mortality [4]. However, in that previous study, the predictive capacity of the P/F ratio adjusted and unadjusted for altitude was not evaluated. Considering this aspect, the researchers hypothesized that the P/F ratio adjusted for altitude would exhibit superior predictive performance for in-hospital mortality compared to the unadjusted P/F ratio. Surprisingly, the results of the current study did not support this hypothesis, as both the adjusted and unadjusted P/F ratios showed similar predictive capabilities. These findings suggest that in patients with ARDS, the ideal cutoff point of the P/F ratio to establish a vital prognosis may not be solely determined by altitude adjustments, and the clinical outcomes of ARDS are influenced by multiple complex factors.

This study has some weaknesses. The first is that the cohort came from a single center, so the findings may not be generalizable to other populations. Second, this study adhered to the

**Table 5. Mortality according to ARDS severity.**

| | Mild ARDS mortality (%) | Moderate ARDS mortality (%) | Severe ARDS mortality (%) | Total mortality (%) |
|---|---|---|---|---|
| Berlin [1] | 27 | 32 | 45 | 34 |
| LUNG SAFE [2] | 34.9 | 40.3 | 46.1 | 40 |
| Caser et al. [29] | 30.6 | 43 | 46 | 43.2 |
| Ferrando et al. [16] | 24 | 29 | 39 | 32 |
| Schuijt et al. [30] | 25.3 | 31.3 | 32 | 31 |
| Rodriguez et al. | 29.7 | 46.3 | 65 | 43.7 |
| Rodriguez et al. adjusted for altitude | 36.2 | 52.6 | 71.2 | 46.5 |

Different studies.

definition of Kigali [24], which adjusts the Berlin definition and validates that the use of PEEP is not required to allow the diagnosis of ARDS. This could partly explain the high mortality in this cohort since patients who could benefit from IMV were not identified. Only patients with COVID-19 were included, which does not allow the results to be generalized to other causes of ARDS.

Despite these limitations, this study has several strengths. The first is the large number of patients recruited, which gave it great internal validity and allowed an adequate sample size for the inclusion and evaluation of the variables. Furthermore, this study examined the P/F adjusted by the Berlin recommendation for cities located more than 1,000 masl.

In conclusion, in this cohort of patients with C-ARDS, the adjustment of the P/F did not change the predictive capacity for hospital mortality. Considering these findings and the lack of clinical evidence for this adjustment, we do not recommend adjusting the P/F in the city of Bogotá at 2,640 masl for diagnosis or for clinical decision-making in patients with C-ARDS. Whether this adjustment should be applied to other causes of ARDS has yet to be determined.

## Supporting information

**S1 File. Supplementary material.**
(DOCX)

**S1 Dataset. P/F-unadjusted model.**
(XLSX)

**S2 Dataset. P/F-adjusted model.**
(XLSX)

## Author Contributions

**Conceptualization:** David Rene Rodriguez Lima, Cristhian Rubio Ramos, Andrés Felipe Yepes Velasco, Leonardo Andrés Gómez Cortes, Darío Isaías Pinilla Rojas, Ángela María Pinzón Rondón, Ángela María Ruíz Sternberg.

**Data curation:** David Rene Rodriguez Lima, Cristhian Rubio Ramos, Andrés Felipe Yepes Velasco, Ángela María Pinzón Rondón, Ángela María Ruíz Sternberg.

**Formal analysis:** David Rene Rodriguez Lima, Ángela María Pinzón Rondón.

**Funding acquisition:** David Rene Rodriguez Lima.

**Investigation:** David Rene Rodriguez Lima, Cristhian Rubio Ramos, Andrés Felipe Yepes Velasco, Darío Isaías Pinilla Rojas.

**Methodology:** David Rene Rodriguez Lima, Leonardo Andrés Gómez Cortes.

**Project administration:** David Rene Rodriguez Lima.

**Resources:** David Rene Rodriguez Lima.

**Software:** David Rene Rodriguez Lima.

**Supervision:** David Rene Rodriguez Lima.

**Validation:** David Rene Rodriguez Lima.

**Visualization:** David Rene Rodriguez Lima.

**Writing – original draft:** David Rene Rodriguez Lima, Cristhian Rubio Ramos, Andrés Felipe Yepes Velasco, Leonardo Andrés Gómez Cortes, Darío Isaías Pinilla Rojas, Ángela María Pinzón Rondón, Ángela María Ruíz Sternberg.

**Writing – review & editing:** David Rene Rodriguez Lima, Cristhian Rubio Ramos, Andrés Felipe Yepes Velasco, Leonardo Andrés Gómez Cortes, Darío Isaías Pinilla Rojas, Ángela María Pinzón Rondón, Ángela María Ruíz Sternberg.

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
