## [Decision Letter · Decision Letter 0]

25 Jul 2023

PONE-D-23-14199Prediction model for in-hospital mortality in patients at high altitudes with ARDS due to COVID-19PLOS ONE

Dear Dr. Rodriguez Lima,

Thank you for submitting your manuscript to PLOS ONE. After careful consideration, we feel that it has merit but does not fully meet PLOS ONE’s publication criteria as it currently stands. Therefore, we invite you to submit a revised version of the manuscript that addresses the points raised during the review process.

We look forward to receiving your revised manuscript.

Kind regards,

Muhammad Tarek Abdel Ghafar, M.D

Academic Editor

PLOS ONE

Journal Requirements:

Reviewers' comments:

Reviewer's Responses to Questions

**Comments to the Author**

1. Is the manuscript technically sound, and do the data support the conclusions?

Reviewer #1: Yes

Reviewer #2: Yes

2. Has the statistical analysis been performed appropriately and rigorously? 

Reviewer #1: Yes

Reviewer #2: Yes

3. Have the authors made all data underlying the findings in their manuscript fully available?

Reviewer #1: Yes

Reviewer #2: Yes

4. Is the manuscript presented in an intelligible fashion and written in standard English?

Reviewer #1: Yes

Reviewer #2: Yes

5. Review Comments to the Author

Reviewer #1: The manuscript addressed the specific situation of high altitude as a possible factor to alter the P/F predicted value for mortality, it did not show a significant difference of the corrected prediction despite another study by the author (https://intjem.biomedcentral.com/articles/10.1186/s12245-022-00426-4) from same hospital and almost same period, showed higher incidence of mortality in high altitude country of Colombia after the initial onset of disease. Can you explain this more clearly in the discussion?

Reviewer #2: Dear authors

Many thanks for your valuable efforts. It was within the global readership interest, well written and done correctly and soundly. Availability of all data and following ethical guidelines were one of the strength points in this article.

6. PLOS authors have the option to publish the peer review history of their article (what does this mean?). If published, this will include your full peer review and any attached files.

Reviewer #1: **Yes: **Dr. Hassan Aref Shabana

Reviewer #2: No

---

## [Author Response · Author response to Decision Letter 0]

26 Jul 2023

Response to Reviewers

Thank you very much for reviewing this manuscript and for your observations.

PONE-D-23-14199

Prediction model for in-hospital mortality in patients at high altitudes with ARDS due to COVID-19

PLOS ONE

Journal Requirements:

and 

All the formats for author names, affiliations, and the body of the text were followed correctly.

All references were reviewed, and no retracted articles were found. An error was identified in the introduction, where references skipped from 7 to 23. This was due to a mistake with the reference manager, which did not add 5 articles. The issue has been corrected, and all references have been added and organized to ensure their proper alignment with the corresponding parts of the text. The references are formatted in Vancouver style, following the journal's recommended style.

Comments to the Author

1. Is the manuscript technically sound, and do the data support the conclusions?

Reviewer #1: Yes

Reviewer #2: Yes

OK

2. Has the statistical analysis been performed appropriately and rigorously? 

Reviewer #1: Yes

Reviewer #2: Yes

OK

3. Have the authors made all data underlying the findings in their manuscript fully available?

Reviewer #1: Yes

Reviewer #2: Yes

OK

4. Is the manuscript presented in an intelligible fashion and written in standard English?

Reviewer #1: Yes

Reviewer #2: Yes

OK

5. Review Comments to the Author

Reviewer #1: The manuscript addressed the specific situation of high altitude as a possible factor to alter the P/F predicted value for mortality, it did not show a significant difference of the corrected prediction despite another study by the author (https://intjem.biomedcentral.com/articles/10.1186/s12245-022-00426-4) from same hospital and almost same period, showed higher incidence of mortality in high altitude country of Colombia after the initial onset of disease. Can you explain this more clearly in the discussion?

In the discussion, the requested point is clarified and highlighted in yellow.

In a study published by the same research group as this work, it was demonstrated that at an altitude of 2640 masl, there exists a significant association between lower P/F ratios and in-hospital mortality[4]. However, in that previous study, the predictive capacity of the P/F ratio adjusted and unadjusted for altitude was not evaluated. Considering this aspect, the researchers hypothesized that the P/F ratio adjusted for altitude would exhibit superior predictive performance for in-hospital mortality compared to the unadjusted P/F ratio. Surprisingly, the results of the current study did not support this hypothesis, as both the adjusted and unadjusted P/F ratios showed similar predictive capabilities. These findings suggest that in patients with ARDS, the ideal cut-off point of the P/F ratio to establish vital prognosis may not be solely determined by altitude adjustments, and clinical outcomes of ARDS are influenced by multiple complex factors.

Reviewer #2: Dear authors

Many thanks for your valuable efforts. It was within the global readership interest, well written and done correctly and soundly. Availability of all data and following ethical guidelines were one of the strength points in this article.

OK

6. PLOS authors have the option to publish the peer review history of their article (what does this mean?). If published, this will include your full peer review and any attached files.

Do you want your identity to be public for this peer review? For information about this choice, including consent withdrawal, please see our Privacy Policy.

Reviewer #1: Yes: Dr. Hassan Aref Shabana

Reviewer #2: No

OK

The PACE tool is used for image adjustment.

---

## [Decision Letter · Decision Letter 1]

14 Aug 2023

PONE-D-23-14199R1Prediction model for in-hospital mortality in patients at high altitudes with ARDS due to

COVID-19PLOS ONE

Dear Dr. Rodriguez Lima,

Thank you for submitting your manuscript to PLOS ONE. After careful consideration, we feel that it has merit but does not fully meet PLOS ONE’s publication criteria as it currently stands. Therefore, we invite you to submit a revised version of the manuscript that addresses the points raised during the review process.

We look forward to receiving your revised manuscript.

Kind regards,

Muhammad Tarek Abdel Ghafar, M.D

Academic Editor

PLOS ONE

Journal Requirements:

Additional Editor Comments:

1- Careful editing for the English language is required by a native English speaker. Please pay attention to the use of abbreviations and provide their full terms at their first mention.

2- Please ensure that all figures, tables, and supplementary tables and figures are cited in the text.

Reviewers' comments:

Reviewer's Responses to Questions

**Comments to the Author**

1. If the authors have adequately addressed your comments raised in a previous round of review and you feel that this manuscript is now acceptable for publication, you may indicate that here to bypass the “Comments to the Author” section, enter your conflict of interest statement in the “Confidential to Editor” section, and submit your "Accept" recommendation.

Reviewer #1: All comments have been addressed

Reviewer #2: All comments have been addressed

2. Is the manuscript technically sound, and do the data support the conclusions?

Reviewer #1: Yes

Reviewer #2: Yes

3. Has the statistical analysis been performed appropriately and rigorously? 

Reviewer #1: Yes

Reviewer #2: Yes

4. Have the authors made all data underlying the findings in their manuscript fully available?

Reviewer #1: Yes

Reviewer #2: (No Response)

5. Is the manuscript presented in an intelligible fashion and written in standard English?

Reviewer #1: Yes

Reviewer #2: Yes

6. Review Comments to the Author

Reviewer #1: Thank you for the clarification added , it is clearly understood and sounds logic

No more questions are required

Reviewer #2: Dear Authors, Many thanks for your response to our review. I hope success for each one of you. Regards

7. PLOS authors have the option to publish the peer review history of their article (what does this mean?). If published, this will include your full peer review and any attached files.

Reviewer #1: No

Reviewer #2: No

---

## [Author Response · Author response to Decision Letter 1]

17 Aug 2023

Rebuttal letter PONE-D-23-14199R1

Dear Dr. Muhammad Tarek Abdel Ghafar, M.D

Thank you for reviewing this document. 

1. Please review your reference list to ensure that it is complete and correct. If you have cited papers that have been retracted, please include the rationale for doing so in the manuscript text or remove these references and replace them with relevant current references. Any changes to the reference list should be mentioned in the rebuttal letter that accompanies your revised manuscript. If you need to cite a retracted article, indicate the article’s retracted status in the References list and also include a citation and full reference for the retraction notice.

a. All references are reviewed, they are complete, all correspond to the correct place, and no retracted articles are found.

2. Careful editing for the English language is required by a native English speaker. Please pay attention to the use of abbreviations and provide their full terms at their first mention.

a. Sent for English editing, certificate attached.

b. The entire document is reviewed. Abbreviations used are linked to the full term in the first mention in the abstract and also in the first mention in the manuscript. This is corrected throughout the text. Additionally, abbreviations that had been previously implemented are removed since they are not significantly repeated throughout the manuscript. This is highlighted in the manuscript using track changes.

3. Please ensure that all figures, tables, and supplementary tables and figures are cited in the text.

a. All figures and tables are cited in the text, and this is highlighted in the manuscript using track changes.

4. Given the editing corrections in the title and abbreviations, the tables and supplementary material are added back again.

5. There are no additional comments from the peer reviewers. We are awaiting any further requirements.

David Rodriguez

Main Author

---

## [Editor Report · Decision Letter 2]

13 Oct 2023

Prediction model for in-hospital mortality in patients at high altitudes with ARDS due to

COVID-19

PONE-D-23-14199R2

Dear Dr. Rodriguez Lima,

We’re pleased to inform you that your manuscript has been judged scientifically suitable for publication and will be formally accepted for publication once it meets all outstanding technical requirements.

Kind regards,

Muhammad Tarek Abdel Ghafar, M.D

Academic Editor

PLOS ONE
---

## [Editor Report · Acceptance letter]

17 Oct 2023

PONE-D-23-14199R2 

Prediction model for in-hospital mortality in patients at high altitudes with ARDS due to COVID-19 

Dear Dr. Rodriguez Lima:

I'm pleased to inform you that your manuscript has been deemed suitable for publication in PLOS ONE. Congratulations! Your manuscript is now with our production department. 

Kind regards, 

on behalf of

Prof Muhammad Tarek Abdel Ghafar 

Academic Editor

PLOS ONE